# Wistar Male Rats (*Rattus norvegicus domestica*) Are Aware of Their Dimensions

**DOI:** 10.3390/ani14233384

**Published:** 2024-11-25

**Authors:** Ivan A. Khvatov, Polina N. Ganza, Alexander N. Kharitonov, Maria V. Samuleeva

**Affiliations:** 1Center for Biopsychological Studies, Moscow Institute of Psychoanalysis, 121170 Moscow, Russia; apollinaria.ganza@gmail.com (P.N.G.); samuleeva@gmail.com (M.V.S.); 2Institute of Psychology, Russian Academy of Sciences, 129366 Moscow, Russia

**Keywords:** self-awareness, body awareness, body size awareness, rats

## Abstract

To control their behavior, animals often need to consider various physical dimensions of their body to predict their interaction with objects in their environment. We studied the ability of brown rats to consider the size of their body. To solve the experimental task, the rats had to penetrate through various holes in the partition of the experimental setup that separated them from the bait. The results of the study showed that rats can choose, from three available holes, the only one that is penetrable even if the other two holes have a larger area. They made their choice even before direct tactile contact with the partition, that is, they anticipated the result of their possible passing through of the holes. These data convincingly show that when solving the task of penetrating holes, rats use the body size awareness mechanism. In turn, this suggests that rats have a certain form of self-awareness.

## 1. Introduction

Body self-awareness is the ability of animals (including humans) to consider the physical parameters of their own body and to anticipate the possible results of interactions with various objects in the environment when planning their behavior [1,2]. Several authors note that this ability is, firstly, the basis for mental regulation of behavior by any animal, and, secondly, it serves as the basis for more complex forms of self-awareness [3,4,5].

The main method used for the experimental study of body self-awareness is the so-called body-as-obstacle task [2,4,5,6,7,8]. The essence of the method is that an animal is faced with a certain task (for example, reaching a bait) when the subject’s own body can be an obstacle to achieving the goal. Accordingly, in order to successfully solve the problem, the animal must consider certain characteristics of its own body. In this case, various body parameters can act as an obstacle, e.g., weight (when body weight awareness is studied) or size (when body size awareness is studied). To date, the body weight awareness has been identified in children aged 22–26 months [6], Asian elephants [1], domestic dogs [7], and brown rats [8]. body size awareness has been found in children aged 22–26 months [6], domestic dogs [5,9] and cats [10], ferrets [2], budgerigars [11], gray crows [12], snakes [13], and crabs [14,15].

This present study is devoted to body size awareness in brown rats. Both in the wild and in synanthropic conditions, rats live in burrows; therefore, penetration into holes is an obvious ecological task for them [16]. In this study, we used an adaptation of a technique that has previously been repeatedly tested and proven its effectiveness with different species of animals [2,12]. In solving the experimental task, the animals were required to pass from one compartment of the setup to another through a partition with three holes, the sizes of which varied. The experiment included two series. In the first series, all holes were passable, but their size and position varied in each trial. We tried to determine whether the animals preferred entering larger or smaller holes. The purpose of the second experimental series was to determine whether subjects could, without prior physical contact, choose a single, smaller, penetrable hole against two larger but impenetrable holes [2]. In order to exclude the learning effect, firstly, the position of the holes in the partition varied quasi-randomly, and secondly, two types of experimental trials were alternated, namely the test and background trials. In the test trials, the dividing partition had one penetrable hole of a smaller size and two impenetrable ones of a larger size. In the background trials, on the contrary, only the small hole was impenetrable, and the two large ones were penetrable. Earlier, the results obtained using this method made it possible to discover the ability of body size awareness in gray crows and ferrets.

Various strains of brown rats are known to be a classic model object in laboratory studies [16]. Their cognitive processes have been well studied. Rats are characterized by nocturnal and twilight activity. The leading modality is olfaction; vision is poorly developed and depends on the strain [16,17,18]. Rats are social animals with a well-developed spatial memory [19,20,21]. Vibrissae play an important role in their lives [22,23]. They allow the animal to determine the depth of space and running speed [24,25]. They are the basis of proprioception [26], protect the eyes [27,28], and allow the animals to determine the size of holes with high accuracy [29]. As for body self-awareness in rodents, the ability of brown rats to take their own body weight into account when determining the strength of a support has been established to date [8], and mice can perform mirror-induced self-directed behavior in the presence of additional tactile stimulation [30].

## 2. Materials and Methods

### 2.1. Subjects

Six male brown rats (Rattus norvegicus) of the Wistar strain, aged 5 to 6 months, participated in the experiment. The weight of the rats at the beginning of the experiment was 400–420 g. All animals were of different litters and were born from different females and males. The animals were raised at the Center for Bio-Psychological Research, Moscow Institute of Psychoanalysis, Moscow, Russia.

The subjects underwent a handling procedure for two weeks before the experiment. All animals were handled by the same person. Individuals were marked at the base of the tail with water-based markers; the marks were renewed during the study.

#### 2.1.1. Experimental Sample Size

Some other studies of body self-awareness have used large experimental samples consisting of dozens of animals [5,9,10,14,15], with each animal participating in only a few experimental trials.

In this study, we tried to clarify whether each animal, in a series of similar but not identical experimental conditions, could make choices that indicate its ability to consider its own body size. Accordingly, we developed an experimental design consisting of many trials (see Section 2.5.1 and Section 2.6.1). Additionally, we controlled for the possibility of learning as an undesired secondary variable (see Section 2.6.2). Due to the length of the experimental series, we used the smallest possible number of individuals—just sufficient to collect data that could be valued statistically.

#### 2.1.2. Animal Keeping Conditions

The experimental animals were kept at a 12/12 light cycle; the air temperature did not exceed 21 degrees Celsius.

The rats were kept in three cages, with two animals in each. The cages were 540 × 390 × 210 mm in size, made of polycarbonate, and covered with metal lattice lids.

The aim of the experiments was to establish the rats’ ability to take their own body size into account. Therefore, we did not enrich their environment with objects with holes that the animals could penetrate. The cages were equipped only with a water bottle, wood pellets (bedding), and wood fiber (nesting material).

#### 2.1.3. Food Deprivation

The rats were fed with complete feed for laboratory animals (supplier: Laboratorkorm, Moscow, Russia).

Twelve hours before the experimental tests, the rats were deprived of food to motivate them. After the completion of the experimental tests on a particular day, the rats were given free access to food. During the entire experiment, all animals lost no more than 15% of their initial weight.

Access to water was around the clock.

All experimental procedures were established in accordance with ARRIVE guide-lines and the European Communities Council Directive for the Care and Use of Laboratory Animals (2010/63/EU) [31], and were approved by the Ethics Committee of the Moscow Institute of Psychoanalysis, where the study was conducted.

### 2.2. Experimental Setup

The experimental setup was a rectangular box 1000 mm long, 600 mm wide, and 500 mm high, divided into four parts by three partitions. The side walls and the floor were made of white opaque organic glass; the partitions and insert panels were made of black opaque organic glass. The “start” and “finish” sections were located on the left and right side. They communicated with the central section by openings 550 mm wide and 250 mm high. These openings could be covered with black opaque organic glass plates inserted from above. In the middle of the central compartment there was a partition with three rectangular openings 150 mm wide and 250 mm high. The size and shape of each opening could be changed by means of additional plates containing openings of different types (see description below), also inserted from above (Figure 1).

The panels with holes inserted into the grooves were plates of black opaque organic glass 600 mm high and 155 mm wide. The holes of various shapes were situated in the lower part of the panel 5 mm from the lower edge (Figure 2 and Figure 3).

Video recording was carried out using a Panasonic HC-V260 video camera (Panasonic Holdings Corp., Petaling Jaya, Selangor, Malaysia) mounted on a tripod 1 m above the central partition.

### 2.3. General Procedure

Throughout the experiment, the animals’ weight was maintained below normal (in total, the animals lost no more than 15% of their initial body weight).

At the beginning of the test, before the animal was placed into the experimental setup, the opening connecting the starting compartment with the others was covered with a plate, and plates with holes of different sizes or without holes (according to the experimental scenario; see the description below) were inserted into three openings in the central partition. The opening connecting the finishing compartment with the others remained open. A bait was placed in the finishing compartment (standard laboratory food for rodents).

The tested rat was then placed in the starting compartment. The experimenter waited 30 s for the animal to calm down and then lifted the panel separating the starting compartment and the rest of the setup. The animal then moved into the finishing compartment, choosing one of the holes in the central partition for entry. As soon as the rat took the bait, the finishing compartment was closed with a panel. The trial was considered complete if the rat entered the finishing box and took the bait within 5 min. If the animal did not enter the finishing box within 5 min, the trial was stopped.

There were 117 trials per animal: 9 training trials, 36 trials in the first experiment, and 72 trials in the second experiment. Each subject performed 10 to 15 trials per day, so the rat subjects were trained for 10 to 12 days.

### 2.4. Training

Before training, each rat was placed in the experimental setup with all holes open (without inserts) and was allowed to move around freely for 20 min. This was carried out to ensure that the animal became used to the experimental setup.

The training series was conducted to familiarize the subjects with the experimental procedure. It consisted of nine trials for each animal. In each trial, one of the holes was completely open (without an insert). The trials were alternated quasi-randomly, observing two conditions. First, in each subsequent trial, the hole was opened in a new position (left, center, or right), relative to the previous trial. Second, the hole in each position (left, center, or right) was opened three times.

Our previous similar experiment with ferrets showed that such familiarization training is sufficient for the animals to become accustomed to both the experimental setup and the experimental procedure [2].

The training was considered completed when a rat reached the bait in no more than five seconds using the shortest way. By the last two trials, the time it took all subjects to solve the problem was 3–5 s.

### 2.5. Experiment 1

#### 2.5.1. Experimental Procedure

The purpose of this experiment was to determine whether the rats would prefer larger holes when reaching for bait.

In pilot trials, we found that the largest rat could penetrate a circular hole 45 mm in diameter, although not without some difficulties. On the basis of these data, we set the minimum size of the hole that the test animals could penetrate, namely a circle 50 mm in diameter.

The experimental series consisted of 36 trials. In each trial, the central partition contained three round penetrable holes of different diameters: small (50 mm), medium (55 mm), and large (60 mm) (Figure 2). The position of the holes in the partition (left, center, or right) in each trial was set quasi-randomly, depending on the following two conditions. First, during the entire series, a hole of each size had to be in each of the three positions in the central partition (left, center, or right) the same number of times, namely 12. Second, in each subsequent trial, a hole of a certain size had to be in a different position (left, center, or right) than in the previous one.

**Figure 2 animals-14-03384-f002:**
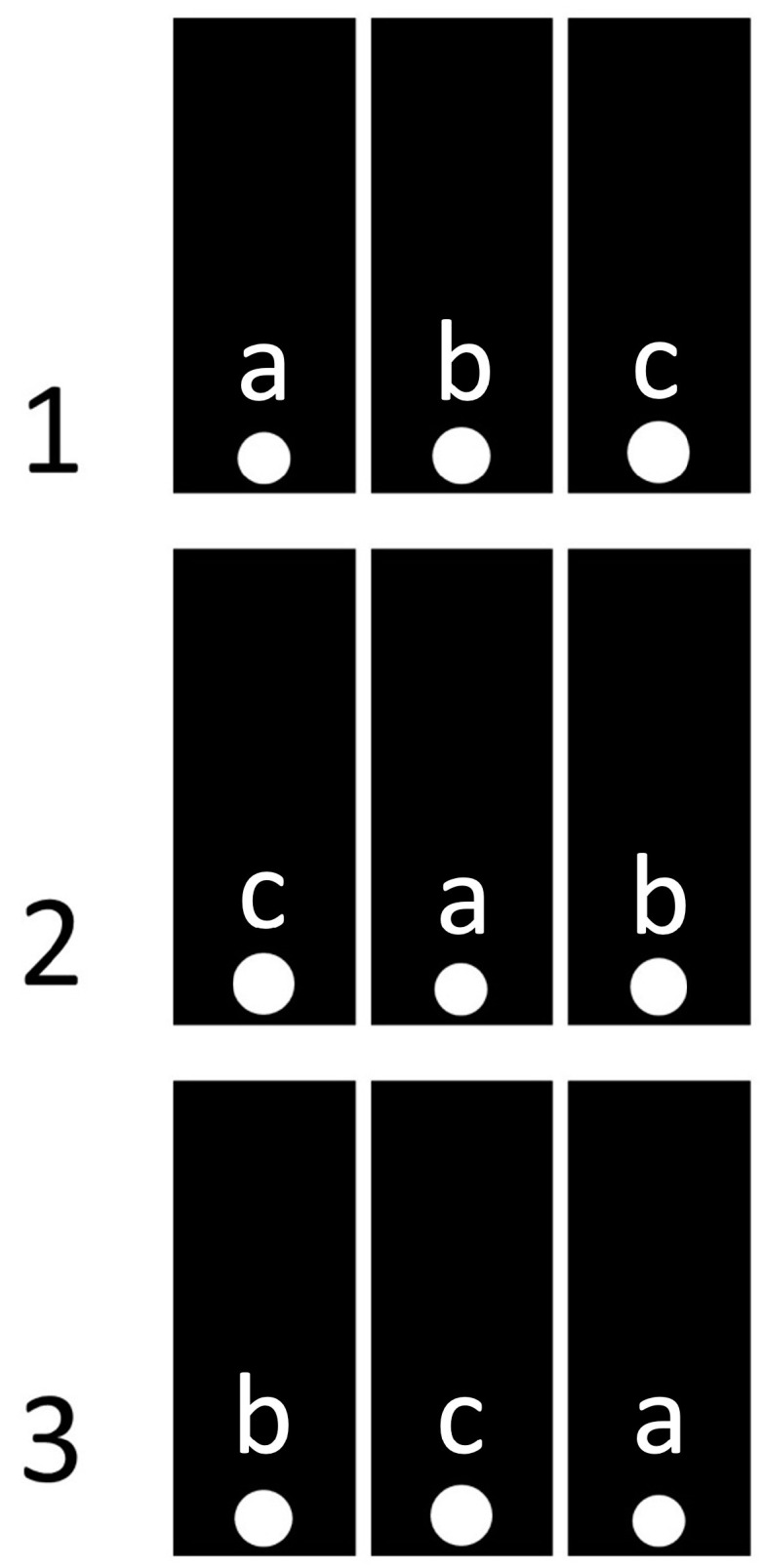
Positions of the inserted panels with holes in the first three trials in the first experimental series. The experimental trials are indicated by numbers. The types and sizes of the round holes are indicated by letters: a—small (Ø 50 mm); b—medium (Ø 55 mm); c—large (Ø 60 mm).

We set the length of the experimental series to 36 trials so that each of the six possible combinations of holes (size and position) was used the same number of times. The same experimental algorithm (and the same number of trials) in a similar experiment was previously used on ferrets and showed its effectiveness [2]. 

#### 2.5.2. Recorded Variables and Statistical Analysis

We recorded the quantitative expression of three dependent variables.

The first approach to a certain opening in each trial. We considered a situation to be a “first approach” when the animal’s muzzle came close to a certain opening so that the distance between the tip of the nose and the partition was no more than 15 mm while the vibrissae did not touch the partition.The first attempt to penetrate a certain hole in each trial. We considered a situation to be “an attempt” to penetrate when a part of the animal’s muzzle penetrated the hole and looked out from the other side of the partition.Penetration into a certain hole carried out in each trial. A situation was considered to be “penetration” when the animal’s whole body passed through one of the holes (but for the tip of the tail). It may be added here that after this, the rat rushed to the bait in 100% of the trials.

We calculated the correlation between the first approach to a certain hole and the subsequent attempt to penetrate that hole. For this purpose, using Pearson’s chi-square criterion (χ^2^), the empirical distribution of the total number (for 6 subjects for all 36 trials) of approaches followed by an attempt to penetrate the hole and the total number of approaches after which the rat attempted to penetrate another hole was compared with a uniform hypothetical distribution (50%/50%).

To identify the factors influencing the rat’s choice of the hole, we used factorial ANOVA (n = 6). The predictor variables were the position of the hole in the partition and the size of the hole. The dependent variable was the number of attempts to penetrate the selected holes. The subject identifier acted as a random factor. In addition, two-way relationships between predictors were considered. To establish differences between the levels of predictors, we used Tukey’s post hoc test.

The data were processed using STATISTICA, version 10.

### 2.6. Experiment 2

#### 2.6.1. Experimental Procedure

The second experiment was conducted to evaluate the ability of rats to choose the only passable hole out of the three holes in the partition, given that the other two would be larger.

The experimental series consisted of 72 trials of two types differentiated according to the criterion of the shapes of the holes used in them, i.e., 24 test trials and 48 background trials.

In each test trial, the partition had one small passable hole and two impenetrable ones which were larger in area. The minimum size of the passable hole was based on the empirically established ability of the largest rat, as noted above, to barely penetrate a circular hole of 45 mm in diameter.

In the test trials, we used small passable holes of two types.

A circle with a diameter of 50 mm (Figure 3, b1);A square with a side of 50 mm (Figure 3, b2).

We also used large impenetrable holes of two types in the test trials (their area was 152% of the area of the 50 mm circle).

A horizontal rectangle with a width of 150 mm and a height of 20 mm (Figure 3, a3);A vertical rectangle with a width of 20 mm and a height of 150 mm (Figure 3, a4).

In each test trial, the types of combinations of holes in the partition alternated, creating the following conditions. First, in each subsequent test trial, the small penetrable hole occupied a new position. Second, in the test trials, the shapes of penetrable and impenetrable holes alternated, as well as their combinations, so that in 6 trials, a circle was combined with horizontal rectangles; in 6 trials, a circle was combined with vertical rectangles; in 6 trials, a square was combined with horizontal rectangles; and in the other 6 trials, a square was combined with vertical rectangles. Such alternations were made to exclude the formation of the skill of choosing a hole on the basis of its shape or its combination with holes of other types.

In each background trial, two panels with large penetrable holes and one with a small impenetrable hole were installed in the partition.

In the background trials, we used two types of small impenetrable holes:A circle with a diameter of 20 mm (Figure 3, a1);A square with a side of 20 mm (Figure 3, a2).

We also used large penetrable holes of two types in the background samples:A horizontal rectangle with a width of 150 mm and a height of 50 mm (Figure 3, b3);A vertical rectangle with a width of 50 mm and a height of 150 mm (Figure 3, b4).

**Figure 3 animals-14-03384-f003:**
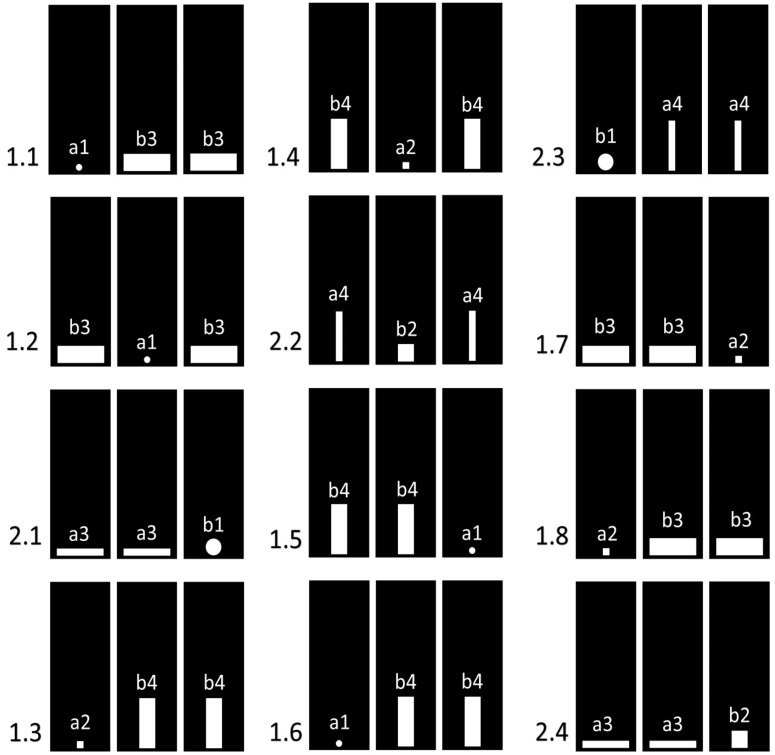
Positions of the panels with holes in the first 12 trials of the second experimental series. The numbers on the left indicate the experimental samples: 1.n—background trials; 2.n—test trials. The letters with numbers in the panels indicate the types of holes: a—impenetrable; b—penetrable. For explanations, see Section 2.6.1.

The types of combinations of holes in the partition in each background trial alternated, using the following conditions. First, in each subsequent trial, the small impenetrable hole had to occupy a new position. Second, the shapes of the penetrable and impenetrable holes, as well as their combinations, alternated in such a way that in 12 trials, the circle was combined with horizontal rectangles; in 12 trials, the circle was combined with vertical rectangles; in 12 trials, the square was combined with horizontal rectangles; and in the other 12 trials, the square was combined with vertical rectangles.

The test and background trials in the experimental series alternated in such a way that two background trials were followed by one test trial. The experimental series began with background trials. Such alternation was necessary to exclude the formation of the habit of penetrating either the smallest hole or the largest hole (in comparison with the other two). 

We used 24 experimental test trials so that each of the 12 possible combinations of holes (shape of penetrable holes, shape of impenetrable holes, hole position) was used the same number of times. Accordingly, the number of times each combination was used in the 48 background trials was twice as large. Moreover, exactly this number of trials demonstrated effectiveness in a study on ferrets using a similar technique [2].

#### 2.6.2. Recorded Variables and Statistical Analysis

We recorded the quantitative expression of three dependent variables: the first approach to a certain hole in each trial, the first attempt to penetrate a certain hole in each trial, and the facts of penetration into certain holes (for a detailed description of the behavioral patterns; see Section 2.5). As in the previous series, after passing through the hole, all rats rushed to the bait.

We established the presence of a relationship between the first approach to a certain hole and the subsequent attempt to penetrate the same hole separately for 24 test and 48 background trials. For this purpose, using Pearson’s chi-square criterion (χ^2^), the empirical distribution of the total number (for 6 subjects in all test or background trials) of approaches followed by an attempt to penetrate into the same hole and the total number of approaches after which the rat attempted to penetrate another hole was compared with a uniform hypothetical distribution (50%/50%).

Additionally, we assessed the frequency of occurrence of the first approaches to the penetrable hole in each trial with subsequent penetration into the same hole in test trials for all six rats. For this purpose, using Pearson’s chi-square criterion (χ^2^), we compared the empirical distribution of the ratio of the number of these patterns and the number of first approaches to impenetrable holes during 144 trials with a hypothetical uniform distribution, where the probability of approaching any hole was 1/3.

We also determined in how many cases out of 24 test trials each animal made the first approach with a subsequent attempt and penetration into the penetrable hole. For this purpose, a binomial test was used, i.e., the actual number of first approaches to the penetrable hole with subsequent penetration into the same hole for each individual was compared with a uniform value (considering that only one of the three holes in each trial was penetrable, it was 33.3%). In addition, we established the effect size using Cohen’s “d” [32].

To identify the factors influencing the rat’s choice of the hole for the first approach in each trial, as well as the first penetration attempt in each trial, for 24 test trials, we used factorial ANOVA (n = 6). The predictor variables were (a) the penetrability or impenetrability of the hole, (b) the position of the hole in the partition, (c) the shape of the small penetrable hole, and (d) the shape of the large impenetrable hole. The dependent variables were the first approaches to certain holes in each trial and the first penetration attempts into certain holes in each trial. The subject identifier served as a random factor. In addition, two-way interactions between predictors were considered. To establish differences between the levels of predictors, we used Tukey’s post hoc test.

In the same way, to identify the factors that influenced the rat’s choice of the hole for the first approach in a trial, as well as the first penetration attempt in a trial, 48 background trials were analyzed using factorial ANOVA (n = 6).

Cochran’s Q test was used to determine whether the number of errors (attempts to penetrate impenetrable holes) changed over the 24 test trials.

In addition, the behavior of rats after the first approach to the impenetrable hole in the trial was analyzed for 24 test trials. In order to find out whether rats used the “change hole” strategy after the first approach to the impenetrable hole, we compared the empirical distribution of the number of approaches to the impenetrable hole, after which the rat immediately went to another hole, and the cases when rats immediately attempted to penetrate this impenetrable hole, with a hypothetical uniform distribution (50%/50%). For comparison we used Pearson’s chi-square criterion (χ^2^).

## 3. Results

### 3.1. Experiment 1

In most experimental trials, all six animals made their first attempt to penetrate through the same hole they first approached, i.e., 207 of 216 trials (χ^2^ = 114.884; df = 1; *p* < 0.001). In all 207 cases, the attempt to penetrate ended with the animal’s actual penetration through that same hole and subsequently reaching the bait.

Size was not a predictor of hole choice (F_(2, 45)_ = 1.08; *p* > 0.1). Consequently, the rats did not prefer larger holes.

The position of the hole in the dividing partition served as a predictor of the choice of the hole for penetration (F_(2, 45)_ = 44.26; *p* < 0.001) (Table 1, Figure 4, Appendix A). The rats avoided the central hole regardless of its size, and preferred the left (Tukey’s test, *p* < 0.01) or right (Tukey’s test, *p* < 0.01) hole for penetration.

### 3.2. Experiment 2

In most test trials, all six rats made their first attempt to penetrate through the same hole they had approached first, i.e., 104 out of 144 cases (χ^2^ = 12.779; df = 1; *p* < 0.001). In 97 of the 104 cases, the attempt to penetrate resulted in actual penetration of the hole; in 7 cases, the rats attempted to penetrate into an impenetrable hole. During the test trials, rats significantly more often made their first approach to a penetrable small hole, with a subsequent attempt to penetrate into it, than first approaches to impenetrable large holes (97 of 144 cases; χ^2^ = 33.349; df = 1; *p* < 0.001).

Analysis of the individual behavior of individual animals also showed that the frequency of occurrence of the first approaches to the penetrable opening in each trial with subsequent penetration into it reliably exceeds the uniform distribution, i.e., two rats in 15 of 24 trials (*p* < 0.01; binomial test), one rat in 16 of 24 trials (*p* < 0.001; binomial test), and three rats in 17 of 24 trials (*p* < 0.001; binomial test). The effect size for six subjects was remarkably high (Cohen’s d = 8.33; n = 6; M = 16.16, SD = 0.98).

In 24 test trials, the predictor of hole choice for both the first approach (F_(1, 129)_ = 51.45; *p* < 0.001) (Table 1, Figure 5a, Appendix A) and the first attempt at penetration (F_(1, 129)_ = 1345.7; *p* < 0.001) (Table 1, Figure 5b, Appendix A) was hole penetrability. Regardless of other predictors, in most test trials, rats made their first in the trial approach to the hole that was penetrable (Tukey’s test, *p* < 0.01), and in most trials, they made their first penetration into the penetrable hole (Tukey’s test, *p* < 0.01). We would like to emphasize that, unlike in the first experiment, here, the position of the hole was not a predictor of either the choice of the hole for the first approach (F_(2, 129)_ = 0.25; *p* > 0.1) or the choice of the first hole for penetration (F_(2, 129)_ = 0.96; *p* > 0.1).

Over 24 test trials, the number of attempts to penetrate an impassable hole for all six rats was 7 (out of 144 trials) and did not change significantly over the entire series (Cochran’s Q = 17.43, n = 6, df = 23, *p* = 0.787).

In most background trials (Appendix A), all six rats made their first attempt to penetrate through the same hole they first approached, i.e., 234 of 288 trials (χ^2^ = 62.338; df = 1; *p* < 0.001). In 54 cases, rats made their first attempt to penetrate the small impenetrable hole, but then penetrated into one of the large holes, i.e., in a total of 288 trials, not a single rat made an attempt to penetrate into the impenetrable hole; in other words, they did not make a single error. Accordingly, when performing the ANOVA analysis, we used the first approaches to the hole as the dependent variable. The predictor of hole choice was its penetrability (F_(1, 129)_ = 119.14; *p* < 0.001). Invariant with respect to other predictors, rats more often made the first approach and immediately made the first attempt to penetrate the penetrable holes (Tukey’s test, *p* < 0.001).

We also estimated the behavior of rats after the first approach to the impenetrable hole in a total of 24 test trials. According to this criterion, all six rats made a total of 47 errors; that is, in 47 trials, they made the first approach to the impenetrable hole, while in 40 cases, there was no subsequent attempt to penetrate this hole, so they immediately approached another one, and in 7 cases, they made an attempt to penetrate the same hole (χ^2^ = 13.111; df = 1; *p* < 0.001).

## 4. Discussion

### 4.1. Experiment 1

The choice of the hole for penetration was influenced not by its size but by its position in the partition. The rats avoided the central hole, preferring to penetrate through the left or right one. This fact can be explained, firstly, in terms of labor intensity for penetration. As noted earlier, penetration through a round hole with a diameter of 45 mm caused extra efforts by the largest animal as it had to squeeze through. The animals passed through holes with a diameter of 50 mm, 55 mm, and 60 mm without difficulty. Previously, a similar situation was created for experiments on ferrets [2]. The type of task involving penetration into holes is more ecologically valid for ferrets [33,34] and rats [16,17] than, for example, for crows [12].

Secondly, the central hole in the partition was in the very middle of the experimental setup, away from the side walls. Rats tend to avoid being in open space [16,17]. In a similar experiment, unlike rats, ferrets more often penetrated the central hole, which we then explained by the choice of the shortest path to reach the bait [2].

### 4.2. Experiment 2

The choice of a hole for the first approach in a trial and for the subsequent attempt at penetration was influenced exclusively by the penetrability of the hole. We emphasize once again that, unlike the first experiment, the position of the hole did not influence the choice of either the first hole for approach or the first hole for penetration.

In 97 of 144 test trials, all six rats made their first approach and subsequent penetration immediately into the only smaller but penetrable hole. In addition, attention should be paid to the analysis of the individual behavior of the animals. It showed that each of the six rats did not randomly choose the hole of the appropriate size for penetration.

It is also necessary to investigate the possibility that the rats quickly learned the “similar–different” rule and in each new test trial, they chose a hole that was different from the one they had entered previously. However, statistical analysis showed that the number of unsuccessful attempts to pass through impassable but large holes did not change significantly over 24 test trials. In addition, in the background trials, the rats were significantly more likely to make the first approach and subsequently enter one of the large holes (which were identical to each other) but not the small impassable hole. This signals the absence of a learning effect.

The data obtained indicate that rats, solving the problem set before them within the framework of this experiment, were able to choose for their penetration the only suitable hole out of three possible ones regardless of how the other two impenetrable holes had a larger area. Evidently, the rats made their choice even before direct physical contact with the partition with the holes. Accordingly, it should be concluded that rats are able to consider the size of their own body and compare it with the qualities of surrounding objects, i.e., they have body size awareness.

### 4.3. General

The results obtained in the above experiments may be considered in comparison with the data on the features of body size awareness in other animal species. Studies conducted on crows [12] and ferrets [2] are the first candidates for such a comparison, since they used experimental methods almost identical to what is described here. 

In the experiment on crows [12], only three of six birds demonstrated signs of body size awareness according to the criterion of the first approach and subsequent penetration into the penetrable hole. In addition, in crows, in an experimental series similar to our second series, the predictor of the choice of the hole for the first approach was not only its penetrability but also its shape. In trials where impenetrable holes of a larger area exceeded the small penetrable one in height, the birds more often approached the impenetrable but high holes. In those trials where the impenetrable hole exceeded the penetrable one in width, the crows more often made the first approach to the penetrable hole.

In addition, the results of the second experiment suggest that when the problem of choice of a hole causes difficulty, rats use a “loose-shift” strategy. If the first hole they approached turns out to be impenetrable, rats most often move to another one (in 40 cases out of 47) but do not try to enter this hole. Crows demonstrated a different behavior, which is more correctly characterized as “loose-repeat”, i.e., if the first hole they approached turned out to be impenetrable, the birds more often made several attempts to enter it in a row [12]. Therefore, rodents demonstrate greater flexibility in their behavior when solving this problem than corvids.

When comparing the results obtained on crows and rats, one should take significant differences in the general body morphology, physiology of sensory systems, behavior, and ecology of these species into account. In particular, the ability to consider one’s own dimensions is more relevant for rats, given that in natural conditions, they constantly penetrate into various openings [16,17]. Nevertheless, we can state that rats cope better with the task of choosing a hole of suitable size for penetration so far as they make fewer mistakes under similar conditions.

Using the same methodology of experimental research, the phenomenon of body size awareness has been previously studied in domestic ferrets [2]. The results obtained with the ferrets (an equal number, six animals, participated in the experiment) were similar to those we obtained in this study. However, unlike rats, ferrets made fewer errors in choosing a hole to penetrate during the second experiment (according to the criterion of the first approach to the hole). Ferrets made 7 errors in 144 test trials, while rats made 47 errors in 144 test trials. On the other hand, according to the criterion of the first penetration into the holes, the differences are not so significant, i.e., the ferrets made 4 errors in 144 test trials, whereas the rats made 7 errors in 144 test trials. As the task of entering holes of different sizes and shapes is ecologically valid for both ferrets [32] and rats [16,17], we suggest that the differences in the results presented above are explained by the peculiarities of the animals’ vision. When choosing a hole to penetrate even before physical contact with it (according to the criterion of the first approach to the hole), the animals relied on the visual modality. In the experiments described here, we used Wistar rats. Due to albinism, their ability to visually distinguish the shapes of objects is significantly worse than that of rats without albinism, as well as other mammals [18]. On the other hand, rats have a fine kinesthetic differential sensitivity, particularly when determining the size of holes [29]. In this regard, we believe that sometimes the rats made a decision about the penetrability/impenetrability of the hole directly upon physical contact of their vibrissae with the edges of the hole. This explanation is also supported by the “loose-shift” tactic used by the rats, as indicated above.

Other body size awareness studies have used methods that differ significantly from ours, making it difficult to directly compare the results. However, we believe it is important to discuss these studies as well to provide a more complete picture.

In a study of body size awareness in toddlers aged 18–26 months, the subjects also had to choose between two holes, the one that was penetrable to their bodies and the one that was not [6]. Fifty-seven children took part in the experiment, and two trials of penetration into the hole were allocated to each child. The results showed that only at 26 months did children make no errors in choosing the appropriate hole without prior physical contact. Therefore, rats seem to be comparable with 26-month-old children in terms of their ability to take their own body size into account.

In the experiments conducted on parrots [11], domestic dogs [5], and domestic cats [10], the animals interacted with a partition with only one hole, the size of which varied from series to series of experimental trials. Thus, unlike in the experiment described here, the animals did not have to choose a hole but only to make a decision to penetrate a specific hole or not. The dependent variables were folding of the wings in flight through the hole in parrots [11], the speed of decision-making about penetration of the hole in dogs [5], and the presence of a specific “hesitation behavior” when approaching and passing through the hole in cats [10].

A notable feature of these experiments is that both cats [10] and dogs [5] approached the low holes more slowly and/or with greater hesitation than the narrow ones. Similar findings, as we noted earlier, were obtained with crows. On the other hand, both ferrets and rats showed no difference in their behavior when choosing holes of different heights or widths. This difference is probably due to the morphology and ecology of specific animal species, and further research is needed to clarify it.

The above thesis is supported by the results of another study on body size awareness conducted on dogs [9]. The animals had to choose between the shortest route (which involved penetrating a hole) and a long detour. The results showed that dogs can take the size of their own body into account. The shape of the animal’s head also influenced the decision, which the authors explained as the difference in the dogs’ peripheral vision.

It is also important to consider that body size awareness is also demonstrated by cold-blooded animals, both vertebrates and invertebrates. In particular, the radiated ratsnake (*Elaphe radiata*) can learn to recognize holes of certain sizes as impenetrable through trial and error [13]. Hermit crabs (*Coenobita compressus*) can compare their own body size with the dimensions of their shells [14,15]. These facts support the idea that body size awareness is a fundamental ability that is likely to be present in all animals [4].

### 4.4. Limitations of the Study

Our study is exploratory. Accordingly, it was important to demonstrate that the ability to consider one’s own body size is present in representatives of the species *Rattus norvegicus*. However, some limitations of our study should be pointed out.

Despite the fact that all the animals tested demonstrated the ability to take their own body size into account in a series of experimental tests, it should be emphasized that the experimental sample consisted of only six individuals. To establish how typical (common) the discussed ability is in general among representatives of the species, additional studies should be conducted on a larger sample of subjects.

Identifying intersexual differences in body size awareness was not the purpose of our study. In our institute’s laboratory population, by the time the experiment began, the smallest variation in size and weight was found between males. Therefore, we decided to use males to reduce the variability in the body sizes of the test animals and to avoid the procedure of adjusting the holes’ sizes to each individual. However, some recent data show that female behavior is no more variable than male behavior [35,36]. On the other hand, there are differences between male and female rats, i.e., males demonstrate higher spatial ability [37], female Wistar rats show better vertical exploration [38], and sex differences in processing rewards and punishments [39] have also been found. Consequently, it is incorrect to straightforwardly extrapolate our data to female rats. Sexual differences in body size awareness in rats should be clarified in further studies.

## 5. Conclusions

On the basis of the experiments, one may conclude that the ability of rats to consider the size of their own body when interacting with objects in the environment was established, which is a sign of body size awareness. Consequently, one might admit the following.

First, the list of species demonstrating signs of body size awareness should be expanded to include humans starting from the age of 26 months [6], domestic dogs [5,9], domestic cats [10], domestic ferrets [2], hooded crows [12], budgerigars [11], brown rats, and, most probably, radiated ratsnakes [13] and hermit crabs [14,15], which can also consider the size of their body.

Secondly, the successful application of the method we developed in general testifies to the power of the “body-as-obstacle task” method [2,5,6]. In particular, this method makes it possible to identify signs of body size awareness in animals of species that previously failed the mirror self-recognition task [40].

Thirdly, there is a theory assuming that self-awareness emerged quite early in the evolution of the mind and then developed gradually [3,4]. In our opinion, our data provide grounds for assuming that body self-awareness may be the earliest form of self-awareness. On the other hand, self-awareness is probably not a monolithic phenomenon but consists of many relatively autonomous abilities, such as, for example, body weight awareness and body size awareness [2,5], and probably some others that may be based on several mechanisms developed at different evolutionary stages or sometimes learned. Each of the abilities may be better or worse developed in representatives of different animal species, in accordance with the morphology of their effector organs and sensory systems, as well as their ecology. One may also consider individual differences. 

These considerations outline the horizons for further research.

## Figures and Tables

**Figure 1 animals-14-03384-f001:**
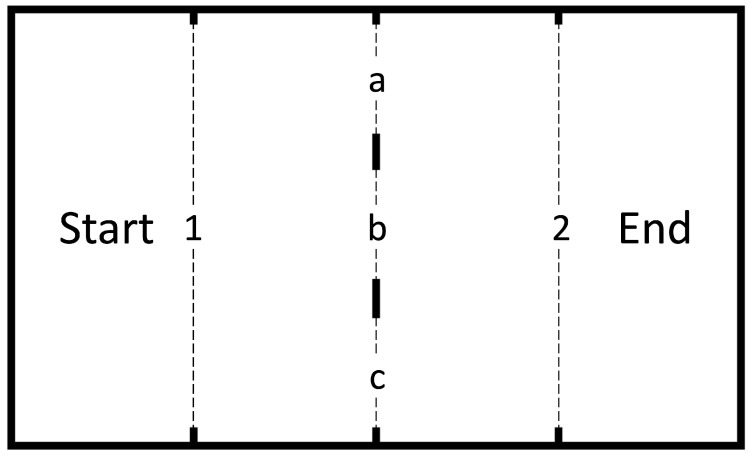
The experimental setup is shown as viewed from above. The dotted lines indicate the openings that could be closed using vertically inserted panels. “Start” is the starting section; “End” is the finishing section; 1 and 2 are the openings that connected the side sections with the central section; and a, b and c are the openings in the central partition, the size and shape of which were varied during the experiment using vertically inserted panels.

**Figure 4 animals-14-03384-f004:**
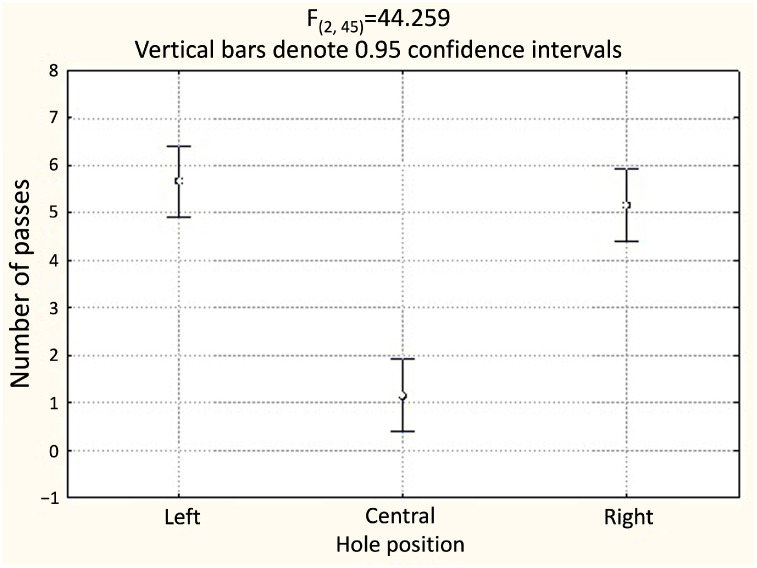
Experiment 1 (n = 6): The average number of passages through the left, central, and right openings in the partition.

**Figure 5 animals-14-03384-f005:**
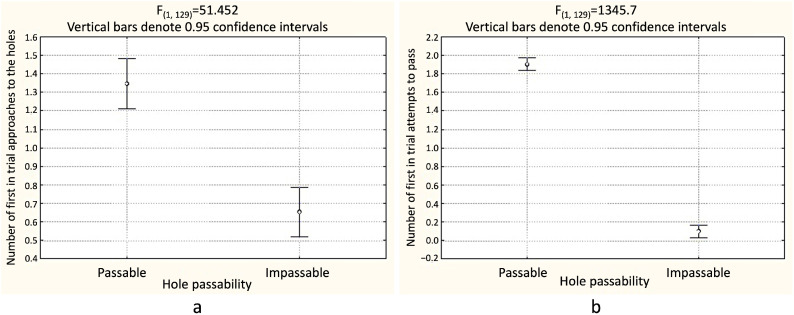
Experiment 2 (*n* = 6): (**a**) the average number of the first approaches in the trial to passable and impassable holes in the partition; (**b**) the average number of first penetration attempts in the trial through passable and impassable holes in the partition.

**Table 1 animals-14-03384-t001:** Significant correlations between the predictors and dependent variables revealed by factorial ANOVA.

Predictor	SS	df	MS	F	*p*
Experiment 1 *					
Hole position	219.0000	2	109.5000	44.2590	0.0001
Experiment 2 ** Number of first attempts in a trial to penetrate through holes					
Hole penetrability	117.3611	1	117.3611	1345.741	0.0001
Experiment 2 ** Number of first approaches to holes					
Hole penetrability	17.3611	1	17.3611	51.4518	0.0001

* Experiment 1. Dependent variable: number of passes through holes. ** Experiment 2, test trials. Dependent variables: number of first attempts in a trial to penetrate holes and the number of first approaches to holes in trials.

## Data Availability

To obtain the data, please contact the corresponding authors.

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
