# Peer review of "Wistar Male Rats (Rattus norvegicus domestica) Are Aware of Their Dimensions"

_animals, 2024, doi:10.3390/ani14233384_

Round 1

Reviewer 1 Report

Comments and Suggestions for Authors

- Introduction: 

line 62, please include quotations

- Methods:

- line 82: did these six rats belong to the same original litter ? Namely, were they siblings, or completely independent from each other ? Please, provide a complete information.

- line 85: cage dimensions and material must be reported

- standard food pellet for laboratory rodents: at least information of the Company should be provided

Author Response

Thank you for reading the manuscript and your comments.

- Introduction:

>>line 62, please include quotations

Quotations have been added (line 80)

- Methods:

>> line 82: did these six rats belong to the same original litter ? Namely, were they siblings, or completely independent from each other ? Please, provide a complete information.

The six rats were not siblings and were born to different females and males.

The information has been added to manuscript (lines 127 – 128)

>> line 85: cage dimensions and material must be reported

The 540 x 390 x 210 mm cages were made of polycarbonate and were covered with steel lattice lids.

The information has been added to manuscript (lines 147 – 148)

>> standard food pellet for laboratory rodents: at least information of the Company should be provided

The rats were fed with standard complete feed pellets for laboratory rodents; supplier: Laboratorkorm, Moscow, Russia.

The information has been added to manuscript (lines 154 - 155)

Reviewer 2 Report

Comments and Suggestions for Authors

The authors present an original study describing rat behavior when exposed to two different experimental settings to challenge and understand rats body size awareness, by assessing preference of rats to use larger holes when reaching for a bait and straight identification of a passable hole in the presence of larger but impenetrable ones. The article is well written, and provides some degrees of novelty, yet several aspects require further and major justification and improvements.

Comment 1: Lines 82-83: Six male brown rats (Rattus norvegicus) of the Wistar line aged 5 to 6 months participated in the experiment.

Reporting the animal model consistently is crucial for accurate data interpretation and reproducibility. Here, the authors refer to brown rat already from the title. Despite brown rat being one of the common names of Rattus norvegicus, when a specific laboratory rat strain is used, the generalization to brown rat is misleading and should be avoided in scientific papers. This is also reinforced by the mention made by the authors (lines 451-453) that some results observed might be associated with albinism (a trait of the wistar and not of the common brown rat).

Already the title should reflect exactly the model used. Furthermore, it is recommended that when only one sex is analyzed in a study, this is also clearly displayed in the title. Thus, the article title should also be reconsidered consistently and state: Wistar male rats (Rattus norvegicus domestica) are Aware of Their Dimensions.

Comment 2: Lines 82-83: Six male brown rats (Rattus norvegicus) of the Wistar line aged 5 to 6 months participated in the experiment.

Neglecting one sex in a study should be avoided whenever possible, unless investigating sex-related diseases. Remarkably, in behavioral analysis, a higher tendency to exploration and better spatial abilities have been described in male wistar rats, when compared to females (Monfort et al. 2015). Another study showed a better vertical exploration in female wistar rats than males (Colettis et al, 2022). It has also been demonstrated that male and female wistar rats differ in decision-making performances (van der Bos et al, 2012).

Furthermore, assumptions that females are more variable than males, that females must be tested across the estrus cycle, and that inclusion of both sexes increases variability, have been countered by many authors (e.g. Beery, 2018 and Becker et al, 2016), thus further advocating for females inclusion in the study.

Authors are recommended to extend their assessment to females too, and to reconsider submission after resolution of sex bias, to improve the quality of their preliminary data.

If impossible, for any reasons, sex bias must be explicitely disclosed as a major limitation of this study, especially considering the existing differences in cognitive and behavioral performances, and remarkably in decision making too, between males and females wistar rats.  A strong scientific justification for the single sex analysis should also be included.

Please see the following:

Monfort P, Gomez-Gimenez B, Llansola M, Felipo V. Gender differences in spatial learning, synaptic activity, and long-term potentiation in the hippocampus in rats: molecular mechanisms. ACS Chem Neurosci. 2015 Aug 19;6(8):1420-7. doi: 10.1021/acschemneuro.5b00096. Epub 2015 Jul 2. PMID: 26098845.

Colettis NC, Habif M, Oberholzer MV, Filippin F, Jerusalinsky DA. Differences in learning and memory between middle-aged female and male rats. Learn Mem. 2022 Apr 15;29(5):120-125. doi: 10.1101/lm.053578.122. PMID: 35428728; PMCID: PMC9053109.

Beery AK. Inclusion of females does not increase variability in rodent research studies. Curr Opin Behav Sci. 2018 Oct;23:143-149. doi: 10.1016/j.cobeha.2018.06.016. Epub 2018 Aug 2. PMID: 30560152; PMCID: PMC6294461.

Becker JB, Prendergast BJ, Liang JW. Female rats are not more variable than male rats: a meta-analysis of neuroscience studies. Biol Sex Differ. 2016 Jul 26;7:34. doi: 10.1186/s13293-016-0087-5. PMID: 27468347; PMCID: PMC4962440.

van den Bos R, Jolles J, van der Knaap L, Baars A, de Visser L. Male and female Wistar rats differ in decision-making performance in a rodent version of the Iowa Gambling Task. Behav Brain Res. 2012 Oct 1;234(2):375-9. doi: 10.1016/j.bbr.2012.07.015. Epub 2012 Jul 16. PMID: 22814113.

Comment 3: Lines 85-93: please provide details of environmental enrichment provided in the cages.

Comment 4: lines 94-96: We considered such an experimental sample size sufficient to obtain reliable empirical data so far as a similar number of individuals was used previously in self-awareness experiments on crows and ferrets [2].

The topic of sample size in behavioral studies can be controversial, with no “one size fits all" approach, recognizing that lab studies and field studies can have major differences in terms of sample size. With a clear focus on reducing the sample size to the minimum possible, to avoid the waste of animals, still it is necessary to use a sufficient pool of experimental animals to obtain significance. Justification for the choice of sample size (6 males) is given in lines 94-96.

The authors are here self-referring (also neglecting the reference to crows study which is n.12 and should be included explicitly). Other studies involving animals addressing the same research question and mentioned in the conclusions (referred to as n.5, 9, 10, 11, 13, 14 and 15), used bigger sample sizes – some of them also including females (added value).

Accurate description on the rational, justification and scientific soundness of sample size calculation, even if a priori, for this specific species and strain (why only 6 males wistar rats?) must be provided.

Comment 5: Line 415: In the experiment on crows [2]

The correct reference shoul be 12.

Comment 6: Supplementary material

Please note that some words reported in Column A in all the three sheets (excel) are still in Russian: please convert to English.

Author Response

The authors present an original study describing rat behavior when exposed to two different experimental settings to challenge and understand rats body size awareness, by assessing preference of rats to use larger holes when reaching for a bait and straight identification of a passable hole in the presence of larger but impenetrable ones. The article is well written, and provides some degrees of novelty, yet several aspects require further and major justification and improvements.

Taken into consideration. Thank you.

Comment 1.

We have changed the title according to your suggestions.

Thank you!

Comment 2.

Thank you for detailed recommendations and a list of references to the topic.

We used male rats to reduce the variability of the body sizes of the test animals. Given the arguments you provided, we agree that it is premature to extrapolate our data to female rats without conducting additional research. So, we have added section 4.4. "Limitations of the study" (lines 730 – 746) to the text with relevant explanations and references to the works you indicated.

Comment 3.

Information about the environmental conditions provided in the cages has been added to section 2.1.2. “Containment conditions” (lines 144 – 152)

Comment 4.

Our study is exploratory. Accordingly, it was important to demonstrate that the ability to consider one's own body size is present in representatives of the species Rattus norvegicus (this also relates to comment 2). In doing so, we aimed to understand whether the one animal, in a series of similar but not identical experimental conditions, can make choices that indicate its ability to consider its own body size. Accordingly, we developed an experimental design consisting of many trials. Due to the length of the experimental series, we used the smallest possible number of individuals -- just sufficient to collect data that can be valued statistically.

We added section 2.1.1. “Experimental sample size” where we provided relevant explanations (lines 133 – 143). We also pointed to the small sample size as a limitation of our study (lines 734 – 738).

Comment 5.

Corrected. Thank you for attentive reading.

Comment 6.

In the supplementary materials, only the names of the rats were written in Russian. We have transliterated them in Latin characters.

Thank you for reading the manuscript and thoughtful recommendations.

Reviewer 3 Report

Comments and Suggestions for Authors

1.      The manuscript presents an experiment evaluating the ability of brown Norway rats to compare their own body size with the diameter of an opening they intend to go through, which the authors termed body self-awareness. Choosing by rat the appropriate opening in a cage partition having three holes of different size, shape, and position was recorded in six young male, food-restricted Wistar rats going for a bait. Results showed that the size of the opening did not affect their choice, but the animals were choosing the closest hole to the walls or were choosing the only passable opening even without the tactile contact with the cage partition. The authors interpreted the result as evidence of body self-awareness.

2.      The results are new, although the design of the experiment is repeated from the authors previous study on ferrets.

3.      The manuscript has all necessary sections, is well designed and organized, with good illustrations and relevant references.

4.      On the critical side, the text is written with a difficult style. The sentences are convoluted, very wordy with some grammatical mistakes and unconventional, and sometimes inappropriate terminology. It is difficult to follow the text, so I would suggest asking a native English speaker to correct the text, as to the sentence structure, phraseology, articles, and flow. I cannot correct the entire text, but I will list some of these problems in an itemized way, below (point # 6).

5.      I also would like to raise a question for the authors’ consideration related to the interpretation of their results, both in their past and in present paper. I have problem with the term “body self-awareness”. This is to me, not only anthropomorphism, but also misnomer ascribing to the rat a complex cognitive function, which rats do not have. In humans with permanent damage to the visual cortex, there is a well-known phenomenon of blindsight. The patients are blind but the motor system for eye movements and scaling is intact (dorsal visual stream). The patients unconsciously use the visual input to motor control system to guide their hand movements towards an object with appropriate finger scaling (opening the fingers in a proportional way to the size of the object being grasped), even though they are completely blind and cannot consciously see the object. So, the visual information is guiding the behavior (dorsal visual stream) without producing conscious sensation or awareness. Consciousness and awareness are produced only by some parts of the brain (e.g., neocortex) but not by the entire brain. Results of the authors’ study showed that rats can automatically judge the dimensions of the opening before passing through it, but I do not think that this is evidence for self-awareness or even awareness. It is rather an automatic motor response based on visual input, which operates outside of consciousness and awareness. In the intact human brain, however, we may talk about conscious awareness, but I have my doubts that this is a case with rats and other animals, particularly invertebrates. I think that authors should, at least, modify their concluding sentence on page 13, lines 507-509 stating that “this ability is one of the most evolutionarily ancient components of self-awareness”.

6.      To help the authors with improvements of their manuscript, I have itemized, the most important suggestions, below.

A)      Page 1, line 9. I would suggest deleting the word “specifically” in this sentence. It is not needed here.

B)      Page 1, line 12. Reporting about results of the study should be always written in the past tense. So, it should be, “The results of the study showed that…” This comment is also relevant to some other places in the manuscript.

C)     Page 1, line 13. The word “permeable” is a misnomer and it should be changed. It should be “penetrable” – i.e., allowing objects to pass through (or “impenetrable”). The term “permeable” has similar meaning but it relates only to fluids and gases, not solid objects. This should be changed in the entire manuscript.

D)     Page 1, line 15. The use of the word “negotiation” is incorrect. This term is borrowed from a sport jargon. In this sentence, it should be “passing through”. So, the fragment of the sentence would be “…they anticipated the result of their possible passing through the holes…” This comment pertains to the use of this term in the entire manuscript.

E)      Page 1, line 30. The present study pertains to rats only and should not be extended to “different animal species” in the conclusion.

F)      Page 1, line 40 and 76. The word “body” should be from lower case letter.

G)     Page 2, line 59. It would be better saying, “the animals preferred entering larger or smaller holes.”

H)     Page 3, line 101. The phrase should be, “…located on the left and right side,”

I)        Page 4, line 118. It should be “video recording” not “registration”. This should be corrected in other places of the manuscript, for example in the title on page 5, line 177 (Recorded variables) and on page 8, line 264.

J)        Page 4, line 130. I think that the phrase “placed in the finishing compartment” should be “placed in the starting compartment”.

K)      Page 6, line 190. It should be here, “We calculated the correlation between…”

L)      Page 7, line 250. It should be here, “in the rest of 12 trials”.

M)    Page 9, line 314. In this sentence, the phrase “207 of 216 cases” is not clear. What are the cases? I think it should be “trials”.

N)     Page 9, line 320. There is a spelling mistake in the horizontal scale description. It should be “Left”, not “Lefr”.

O)     Page 10, lines 350-352. The numbers in the Table 1 are written not in English way of writing numerals. The decimal point should be point, not a coma. So, for instance, “219.0” not “219,0”. This pertains to all numbers in the table. This is important because comma separates thousands, so 219,000 means two hundred nineteen thousand.

P)      Page 11, line 375-377. The sentence does not have proper structure and is unclear. In addition to that, the phrase “number of movements performed” is puzzling. The authors have not measured the number of movements. Also, the end of the sentence is not clear; “…penetration procedures for different holes used in this series being the same for all”. For all what?

Q)     Page 11, line 415. The authors have stated that “In the experiment on crows [2]…” However, the reference nr [2] is not about crows but about ferrets.

R)      Page 11, lines 423-424. The sentence is not clear and grammatically incorrect. It has two predicates and three verbs: “the data obtained…suggest…causes”. It loses its sense.

Author Response

Thank you for reading and commenting the manuscript.

1-3 Thank you for these remarks.

  1. >> I would suggest asking a native English speaker to correct the text

A Russian-American bilingual has read the corrected text and suggested some minor corrections only.

  1. >> I also would like to raise a question for the authors’ consideration related to the interpretation of their results, both in their past and in present paper. I have problem with the term “body self-awareness”

“Body self-awareness” is an English term. In Russian we use “self-reflection” and “taking into account” or “consideration of one’s own body” which also has undesirable connotations both in Russian and in English. Do animals “consider” or “account of” something?

If we understand correctly, the term “body self-awareness” is rather transparent in English, and it limits self-awareness to body awareness. But “self” is still the problem. What is “self” in relation to rats? Something which is NOT other rats? Or other animals? Or… what? Unfortunately, same refers to “awareness”. And to “consciousness”. What is “conscious sensation”? Perception? Can’t one imagine “unconscious awareness”?  

Seems it is clear that “awareness” or “consciousness” in animals and humans are different. And they are surely different across animal species. But even in humans… Some 15 years ago one of us found about 30 definitions of consciousness in different Russian sources. We suspect, the situation in the English language scientific literature is similar. Yes, there is a connection of vision with the dorsal stream, or consciousness with neocortex. However, birds do not have neocortex. Are they unconscious? Absolutely? Or still a little bit conscious? At least some of them?

What is it like to be a bat?

>> a misnomer

One important aspect of the problem is the language we use. So far as we have no words for similar (but most probably not same) phenomena we study in humans and across different animal species (animals only?), our excuse is metaphor. Maybe a lame excuse, still an excuse. In the long run, we do speak about “memory” in relation to computer – any problem? In any case, we have to use words when we describe people, animals, things, phenomena… and experiments. Sometimes the words that we cannot clearly define.

It is obvious, you raise a fundamental and multifaceted question. And a very important one.

The humble purpose of this article is to describe an experimental work and to present the data we obtained. Seems we may discuss the theoretical problem somewhere else.

>> modify their concluding sentence on page 13, lines 507-509 stating that “this ability is one of the most evolutionarily ancient components of self-awareness”.

We have somewhat modified the “Conclusion” and added certain considerations to the point (lines 760 – 763)

  1. >> To help the authors with improvements of their manuscript, I have itemized, the most important suggestions, below.

Thank you, it is extremely helpful.

A)      Page 1, line 9. I would suggest deleting the word “specifically” in this sentence. It is not needed here.

Agree. Deleted.

B)  Page 1, line 12. Reporting about results of the study should be always written in the past tense. So, it should be, “The results of the study showed that…” This comment is also relevant to some other places in the manuscript.

Corrected here and elsewhere.

C)  Page 1, line 13. The word “permeable” is a misnomer and it should be changed. It should be “penetrable” – i.e., allowing objects to pass through (or “impenetrable”). The term “permeable” has similar meaning but it relates only to fluids and gases, not solid objects. This should be changed in the entire manuscript.

Corrected here and elsewhere (over 40 occasions)

D)  Page 1, line 15. The use of the word “negotiation” is incorrect. This term is borrowed from a sport jargon. In this sentence, it should be “passing through”. So, the fragment of the sentence would be “…they anticipated the result of their possible passing through the holes…” This comment pertains to the use of this term in the entire manuscript.

Corrected here and elsewhere.

E)     Page 1, line 30. The present study pertains to rats only and should not be extended to “different animal species” in the conclusion.

Corrected here. However, some cross-species comparison is possible, in our opinion.

F)      Page 1, line 40 and 76. The word “body” should be from lower case letter.

Corrected.

G)     Page 2, line 59. It would be better saying, “the animals preferred entering larger or smaller holes.”

Corrected.

H)     Page 3, line 101. The phrase should be, “…located on the left and right side,”

Corrected

I)        Page 4, line 118. It should be “video recording” not “registration”. This should be corrected in other places of the manuscript, for example in the title on page 5, line 177 (Recorded variables) and on page 8, line 264.

Corrected here and elsewhere.

J)        Page 4, line 130. I think that the phrase “placed in the finishing compartment” should be “placed in the starting compartment”.

Mistake. Corrected.

K)      Page 6, line 190. It should be here, “We calculated the correlation between…”

Corrected.

L)      Page 7, line 250. It should be here, “in the rest of 12 trials”.

the rest 12 trials (meaning the remaining 12 trials) – corrected.

M)    Page 9, line 314. In this sentence, the phrase “207 of 216 cases” is not clear. What are the cases? I think it should be “trials”.

Corrected.

N)     Page 9, line 320. There is a spelling mistake in the horizontal scale description. It should be “Left”, not “Lefr”.

Corrected.

O)     Page 10, lines 350-352. The numbers in the Table 1 are written not in English way of writing numerals. The decimal point should be point, not a coma. So, for instance, “219.0” not “219,0”. This pertains to all numbers in the table. This is important because comma separates thousands, so 219,000 means two hundred nineteen thousand.

Corrected.

P)      Page 11, line 375-377. The sentence does not have proper structure and is unclear. In addition to that, the phrase “number of movements performed” is puzzling. The authors have not measured the number of movements. Also, the end of the sentence is not clear; “…penetration procedures for different holes used in this series being the same for all”. For all what?

This fragment has been somewhat modified, confusing parts deleted (lines 582 – 584)

Q)     Page 11, line 415. The authors have stated that “In the experiment on crows [2]…” However, the reference nr [2] is not about crows but about ferrets.

Corrected.

R)      Page 11, lines 423-424. The sentence is not clear and grammatically incorrect. It has two predicates and three verbs: “the data obtained…suggest…causes”. It loses its sense.

We have slightly modified wording and punctuation (lines 630 – 658).

The authors greatly appreciate the thorough reading of the text and thoughtful comments by the reviewer.

Reviewer 4 Report

Comments and Suggestions for Authors

I was somewhat undecided about this manuscript. The experimental design doesn't seem completely thought through to me and has some weaknesses. The illustrations are of such poor quality that in some cases I could only guess what was supposed to be depicted. The question posed in experiment 1 is not answered at all in the results. Such points usually lead to a rejection for me. However, there is little work on the subject and I assume that the professional world could still benefit from the results. I would therefore like to give the authors the opportunity to revise the weak points in the manuscript.

Title

The results of the Wistar rats can probably be generalized, but I would still replace brown rats with Wistar rats in the title for accuracy.

Abstract

Line 24: „larger but impenetrable“ : what does this exactly mean? Why does "larger" matter?

Line 25: “The results of the first experiment show that size does not affect the choice of a hole for penetration.” That is a strong generalization. The 3 holes examined do not differ very much in size from each other, so I would not rule out a preference across the board if there are major differences.

Material and Methods

Line 82: should be “strain”

I would have liked some more details about the rats (weight, size) as this could be relevant in the setting. If possible, please provide more information.

6 rats are few. How was the group size calculated?

Why were only males used?

Please give the details about the food deprivation.

I am not happy with the idea of fasting. 15% weight loss is quite drastic. In the sense of refinement this should be avoided. It is incomprehensible to me why this cannot be achieved with treats in such an experimental setup.

Line 94: Citing a publication where the authors performed experiments with no sample size calculation as basis for your sample size here is inappropriate.  

Line 150:  What is the basis for the assumption that the training is also sufficient for rats? What was considered sufficient training? Please specify.

I do not fully understand how the number of test runs was chosen? 36 in the first experiment and 72 in the second?

Line 209: the number of trials should not be n

Regarding the experimental design: why did you choose a setup with 3 openings? It is not very surprising that the rats preferer the ones closer to the walls (as we see thigmotaxis in rats)?

How did you include the fact that the two outer openings are obviously preferred in the evaluation of experiment 2?

Line 256: A mere reference to another publication is insufficient in terms of reproducibility. The procedure  should also be described in detail here.

Wha did you decide to use a combination of rectangles and  round shapes?

Results

Figures should be revised (bad quality).

Experiment 1

As the question was if the rats prefer larger holes, this should be shown in the results section.

Discussion

Line 411: This is not very surprising as the first author is the same in all of them (including the manuscript here).

Author Response

The Authors greatly appreciate thorough reading of the text and thoughtful comments of the Reviewer.

Title.

We have modified the title.

Line 24: >> „larger but impenetrable“ : what does this exactly mean? Why does "larger" matter?

Greater area, which presumably signals better penetrability. We have modified the wording (line 26).

Line 25: >> “The results of the first experiment show that size does not affect the choice of a hole for penetration.” That is a strong generalization. The 3 holes examined do not differ very much in size from each other, so I would not rule out a preference across the board if there are major differences.

Thank you for the comment.

The results of the first experiment showed that rats did not show a preference for the greater hole. We have changed the wording (lines 26 – 27).

Line 82: >>should be “strain”

Corrected (lines 21 and 125).

>>I would have liked some more details about the rats (weight, size) as this could be relevant in the setting. If possible, please provide more information.

More detailed information has been added (line 126 – 128).

>>6 rats are few. How was the group size calculated?

Our study is exploratory. Accordingly, it was important for us to demonstrate the ability to consider one's own body size in representatives of the species Rattus norvegicus. At the same time, our task was to understand whether one and the same animal, in a series of similar but not strictly identical experimental conditions, can make a choice indicating its ability to take into account its body size. Accordingly, we developed an experimental plan consisting of a large number of trials. Due to the length of the experimental series, we used the smallest number of individuals to obtain just the amount of data sufficient for statistical evaluation.

We added section 2.1.1. “Experimental sample size” where we provided relevant explanations (lines 133 – 143). We also indicated that the small sample size was a limitation of our study (lines 734 – 738).

>>Why were only males used?

We used male rats to reduce the variability in body size of the test animals.

Thank you for these comments. Yes, we agree, it may be premature to extrapolate our data to female rats without conducting additional studies. We have added section 4.4. "Limitations of the study" (lines 739 – 746) to the text of the manuscript, where we include some relevant explanations and refer to the works you indicated.

>>Please give the details about the food deprivation.

We have added section 2.1.3. "Food deprivation" where we specify the details of deprivation (lines 153 – 164).

Earlier, during pilot tests, we tried to motivate rats with a treat. It was found that when the rat was not hungry, it would not strive to reach the treat.

Food deprivation was rather mild. In accordance with ARRIVE guidelines, the European Communities Council Directive for the Care and Use of Laboratory Animals (2010/63/EU), a weight loss of 15% was the maximum permissible. In fact, our animals lost significantly less.

Line 94: >>Citing a publication where the authors performed experiments with no sample size calculation as basis for your sample size here is inappropriate.

Citation deleted.

Line 150:  >>What is the basis for the assumption that the training is also sufficient for rats? What was considered sufficient training? Please specify.

Thanks for the comment. We've added clarification (lines 266 – 267).

>>I do not fully understand how the number of test runs was chosen? 36 in the first experiment and 72 in the second?

In the first experiment, we set the length of the experimental series to 36 trials so that each of the six possible combinations of holes (size and position) was used the same number of times. This makes rather reliable statistics in the absence effect of learning.

In the second experiment, we used 24 test experimental trials so that each of the 12 possible combinations of holes (shape of penetrable holes, shape of impenetrable holes, hole position) was used the same number of times. Accordingly, the number of times each combination was used in the 48 background trials was twice as large. This is the combination of trials to obtain the amount data reliably sufficient for further statistical evaluation (Sidorenko, E. Methods for Mathematical Processing in Psychology, 2000; Montgomery, D. Design and analysis of experiments, 2008). On the other hand, a greater number of trials might be complicated with the effect of learning. In addition, increasing the number of trials would have forced us to prolong the food deprivation which could have affected the overall state of the animals and the results of the experiment.

Additions have been made to the text (lines 285 – 286 and lines 407 – 410).

Line 209: >> the number of trials should not be n

Corrected (line 340).

>> Regarding the experimental design: why did you choose a setup with 3 openings? It is not very surprising that the rats preferer the ones closer to the walls (as we see thigmotaxis in rats)?

We used a three-hole setup to reduce the probability of a rat randomly choosing one of the holes (50%/50%) in each trial. On the other hand, using more than three holes leads to distraction that complicate the interpretation of the animals’ behavior.

>>How did you include the fact that the two outer openings are obviously preferred in the evaluation of experiment 2?

In the ANOVA statistical analysis, we included the position of the hole in the partition as one of the predictors. The results of the second experiment showed that the position of the hole did not predict either the choice of the first hole to approach or the choice of the first hole to penetrate.

We have added explanations to the sections “Results” (lines 522 – 525) and  “Discussion” (lines 595 – 597).

>> Why did you decide to use a combination of rectangles and round shapes?

This is an interesting methodical issue.

To prevent possible learning that might influence test results, we alternated test trials with background trials. In addition, the trials were designed so that the test trials were as different from the background trials as possible. In background trials, the larger rectangles were vertically or horizontally located and were passable.

In test trials, the larger rectangles were impassable.

In background trials, vertically or horizontally positioned rectangles of larger area were passable. In test trials, rectangles of larger area were impassable.

The alternation of passable round and square holes in the test trials was also used to prevent the rats from getting used to a certain shape of the holes.

The results of the second experiment showed that the shape did not influence the choice of the first hole in the trial, both in test and background trials.

>>Results

>>Figures should be revised (bad quality).

Improved.

>>Experiment 1

>>As the question was if the rats prefer larger holes, this should be shown in the results section.

Thanks! We have modified the relevant fragment (lines 485 – 486)

>>Line 411: This is not very surprising as the first author is the same in all of them (including the manuscript here).

We would be very happy if much more researchers were to study this topic. Since this is not the case, we are trying to make our own humble contribution.

Many thanks for your efforts to improve the manuscript.

Round 2

Reviewer 2 Report

Comments and Suggestions for Authors

The authors consistently implemented suggestions into the article, and it is particularly commendable the open acknowledgment of the limitations of the study, creating a dedicated section.

Nevertheless, justification for the sample size and type is still unclear. Particularly, the new sentence at line 539 "We used male rats to reduce the variability of the body sizes of the test animals." may lack scientific soundness, per se.

Are authors referring to intra- or inter-sex variability?

If authors refer to a smaller variability in males than in females, it's important to remark that Wistar rat is a very standardized strain. Even when bred at a specific institution generating one local substrain, and lacking genetic stability program, the variability of weight at specific ages for females would not be wider than the variability reported in males; conversely, the weight variability of females tends to be slightly smaller than the males one.

See examples of growth charts: 

Wistar Han® outbred rats: RccHan:WIST

Wistar Rat | Charles River

If authors refer to the weight variability of males vs females instead, it must be further explained in the text under the limitations paragraph, as willingly neglecting females to avoid considering a different size range, in a study considering size awareness, would fall again into sex-bias in experimental design. And then, why males only and not females only instead?

Therefore, using only male rats to reduce variability of body sizes deserves further expansion. If, just as a matter of example, authors wanted to remark the specific availability of males only in a specific age/weight range at their institute, and conversely the unavailability of females of a suitable cohort, or the impossibility or unwillingness to adjust proportionally the hole sizes considering average sizes of males and females respectively, this should be made explicit in the text.

Author Response

Dear reviewer, once again we would like to thank you for your kind attention to our work. In this article we did not set a task to clarify how gender influences the ability under study. The influence of gender and other individual characteristics of an animal is a very interesting topic. We shall consider it as a direction for further research.

Following your recommendation, we modified the wording "We used male rats to reduce the variability of the body sizes of the test animals" to read as follows (lines 739 – 743):

“Identifying intersexual differences in body size awareness was not the purpose of our study. In our Institute’s laboratory population, by the time the experiment began, the smallest variation in size and weight was found between males. Therefore, we decided to use males to reduce the variability of the body sizes of the test animals and to avoid the procedure of adjusting hole sizes to each individual”.
